# Mask, Stitch, and Re-Sample: Enhancing Robustness and Generalizability in Anomaly Detection through Automatic Diffusion Models

Cosmin I. Bercea [* 1 2]   Michael Neumayr [* 1]   Daniel Rueckert [1 3]   Julia A. Schnabel [1 2 4]

## Abstract

The introduction of diffusion models in anomaly detection has paved the way for more effective and accurate pseudo-healthy synthesis. However, the current limitations in controlling noise granularity hinder the ability of diffusion models to generalize across diverse anomaly types and compromise the restoration of healthy tissues. To overcome these challenges, we propose *AutoDDPM*, a novel approach that enhances the robustness of diffusion models. *AutoDDPM* utilizes diffusion models to generate initial likelihood maps of potential anomalies and seamlessly integrates healthy tissues in the de-noising process. By re-sampling from the joint noised distribution, *AutoDDPM* achieves harmonization and in-painting effects. Our study demonstrates the efficacy of *AutoDDPM* in replacing anomalous regions while preserving healthy tissues, considerably surpassing diffusion models' limitations. It also contributes valuable insights and analysis on the limitations of current diffusion models, promoting robust and interpretable anomaly detection in medical imaging — an essential aspect of building autonomous clinical decision systems with higher interpretability.

Code: https://github.com/ci-ber/autoDDPM

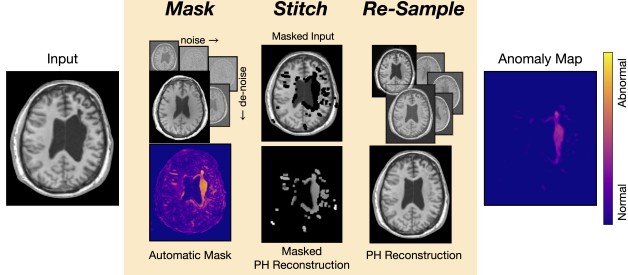

Figure 1. We **automatically** generate initial **anomaly masks** using diffusion models. The unmasked original healthy tissues are then **stitched** to the reconstruction and incorporated into the de-noising process. By **re-sampling** from their joint noised distribution, we achieve harmonization and in-painting effects, resulting in accurate pseudo-healthy (PH) reconstructions and precise anomaly maps.

## 1. Introduction

Anomaly detection involves identifying and characterizing deviations from expected normal patterns or structures. Its application holds immense potential for enhancing diagnostic accuracy and ultimately improving patient outcomes in clinical practice.

*Equal contribution [1]Technical University of Munich, Germany [2]Helmholtz AI and Helmholtz Center Munich, Germany [3]Imperial College London, UK [4]King's College London, UK. Correspondence to: Cosmin I. Bercea <cosmin.bercea@tum.de>.

*Workshop on Interpretable ML in Healthcare at International Conference on Machine Learning (ICML)*, Honolulu, Hawaii, USA. 2023. Copyright 2023 by the author(s).

Unsupervised methods for anomaly detection have gained considerable attention in recent years, capitalizing on the power of large-scale normal data to identify deviations or abnormalities. In contrast to supervised methods, these approaches do not require labeled anomaly samples during training. This makes them more versatile and adaptable to real-world scenarios, as they can generalize well to unseen pathology and diverse clinical settings. In both computer vision and medical domains, extensive research has been conducted on anomaly detection, encompassing techniques such as one-class classifiers (Ruff et al., 2018), knowledge distillation (Bergmann et al., 2020; Salehi et al., 2021), leveraging large nominal-banks (Roth et al., 2022), training on sef-supervised pre-text tasks (Kascenas et al., 2022; Tan et al., 2022), and restoration-based methods (You et al., 2019; Zimmerer et al., 2019; Pawlowski et al., 2018).

Restoration-based methods aim at reconstructing input from a latent representation which encodes features of the healthy distribution. They hold particular significance in the medical field, as they offer valuable interpretability by generating pseudo-healthy (PH) reconstructions that aid clinicians in comprehending the detected anomalies. By reconstructing normal appearance in the absence of anomalies, clinicians can better understand and assess the severity and impact of

detected abnormalities. Traditionally, auto-encoders have been widely used for anomaly detection and reconstruction tasks (Yoon et al., 2021; You et al., 2019). However, they present several challenges, such as difficulties in capturing complex variations, restoring samples with high accuracy from their latent representations, and limited generalization capabilities across different anomaly types and sizes (Bercea et al., 2022). Consequently, recent advancements in the field have turned towards diffusion models as a promising alternative for reconstruction-based approaches.

Diffusion models (Ho et al., 2020) have reignited the interest in reconstruction-based anomaly detection approaches, offering potential solutions to the challenges faced by auto-encoders. These models employ de-noising techniques and provide a more precise framework for anomaly detection and restoration in medical images. Despite their promise, diffusion models still face certain challenges. First, the granularity, level, and type of noise applied in the diffusion process significantly impacts the types of anomalies that can be effectively detected (Wyatt et al., 2022). Typically, in diffusion models, noise is introduced up to an intermediate level rather than complete random noise, aiming to achieve a satisfactory reconstruction of the original image while eliminating anomalies. The careful selection of this noise level is crucial to strike a balance between anomaly removal and preserving sufficient signal for accurate reconstructions. However, different anomaly types and sizes may require distinct noise distributions for effective reconstruction, posing a challenge for generalization across diverse anomalies. Furthermore, the introduction of noise in the diffusion process, while successfully removing anomalies, can also disrupt healthy tissues beyond the possibility of accurate restoration. Thus, finding an optimal trade-off between preserving healthy tissue information and effectively removing anomalies remains a critical and challenging task.

Recent developments in diffusion models have addressed these challenges through various approaches, including adapting the noise distribution to match the target anomaly distribution (Wyatt et al., 2022), employing context information to enhance the de-noising process (Behrendt et al., 2023), exploiting the characteristic of noise selection in diffusion models for out-of-distribution detection (Graham et al., 2022), or incorporating classifier guidance to augment the detection and restoration process (Wolleb et al., 2022). Recently, Bercea et al. (2023b) introduced an unsupervised automatic in-painting pipeline for anomaly detection. Their method involves utilizing latent generative techniques to compute masks, followed by employing generative adversarial networks (GANs) to perform in-painting of pathologies with pseuo-healthy tissues.

In contrast, our proposed method, *AutoDDPM*, offers a novel alternative by utilizing a single diffusion model to seamlessly integrate automatic masking and in-painting tasks, eliminating the need for training additional networks.

The primary objective of our research is to enhance the robustness and generalizability of diffusion models in medical anomaly detection. In summary our contributions are:

- We highlight the limitations of diffusion models, specifically the challenge of selecting an appropriate noise level for detecting stroke lesions of various sizes.

- We introduce a novel approach, termed *AutoDDPM*, which addresses these limitations through the integration of automatic masking, stitching, and re-sampling techniques for anomaly detection.

- We conduct two ablation studies to analyze the impact of re-sampling and leveraging the inherent uncertainty in the initial masking process.

## 2. De-noising Diffusion Probabilistic Models

Denoising Diffusion Probabilistic Models (DDPMs) work in a two-phase process: a forward diffusion procedure $q(x_t|x_{t-1})$ progressively deteriorates data from a target distribution $q(x_0)$ to Gaussian noise $x_T \sim \mathcal{N}(0, 1)$ at timestep $T$. Step by step, the forward process adds Gaussian noise scaled by a variance schedule $\beta_t$ that is set to increase linearly from $\beta_1 = 10^{-4}$ up to $\beta_T = 0.02$ following (Ho et al., 2020). For one step, the process is defined by

$$q(x_t|x_{t-1}) = \mathcal{N}(x_t; \sqrt{1 - \beta_t}x_{t-1}, \beta_t\mathbf{I}). \quad (1)$$

The reverse or denoising process then tries to learn $p_\theta(x_{t-1}|x_t)$ to go back from $x_T \sim \mathcal{N}(0, 1)$ and reverse the deterioration of the forward model. It learns to generate samples of $q(x_0)$ out of noise. There are several possibilities on how to parameterize this generative model. In this work, we estimate $\mu_\theta(x_t, t)$ and fix the variance to $\Sigma_\theta(x_t, t) = \frac{1 - \alpha_{t-1}}{1 - \alpha_t}\beta_t\mathbf{I}$ as reported by Ho et al. (2020). Thus, we derive the reverse process from $T$ to 1 as:

$$p_\theta(x_{t-1}|x_t) = \mathcal{N}(x_{t-1}; \mu_\theta(x_t, t), \tilde{\beta}_t\mathbf{I}), \quad (2)$$

with $\tilde{\beta}_t$ our fixed variance. We can then train a Unet (Ronneberger et al., 2015) to learn the noise $\epsilon_\theta(x_t, t)$ at timestep $t$ and estimate $\mu_\theta$ via:

$$\mu_\theta(x_t, t) = \frac{1}{\sqrt{\alpha_t}}(x_t - \frac{\beta_t}{\sqrt{1 - \bar{\alpha}_t}}\epsilon_\theta(x_t, t)), \quad (3)$$

where $\alpha_t = 1 - \beta_t$ and $\bar{\alpha}_t = \prod_{s=1}^{t} \alpha_s$.

We can leverage this cumulative product expression of the variance schedule up to timestep $t$ to speed up the forward process with the closed form:

$$q(x_t|x_0) = \mathcal{N}(x_t; \sqrt{\bar{\alpha}_t}x_0, (1 - \bar{\alpha}_t)\mathbf{I}). \quad (4)$$

With this formulation, we can efficiently sample training data for random timesteps $t \sim Uniform(1, T)$ and train the model to reverse them.

To get the objective function, we decompose the variational lower bound into three parts:

$$\mathcal{L}_{vlb} = \mathbb{E}_q \left[ \underbrace{\mathbb{D}_{KL}(q(x_T|x_0)\|p(x_T))}_{\mathcal{L}_T} \right. \tag{5}$$

$$+ \sum_{t>1} \underbrace{\mathbb{D}_{KL}(q(x_{t-1}|x_t, x_0)\|p_\theta(x_{t-1}|x_t))}_{\mathcal{L}_{t-1}} \tag{6}$$

$$\left. - \underbrace{\log p_\theta(x_0|x_1)}_{\mathcal{L}_0} \right]. \tag{7}$$

and then apply reformulations and simplifications to arrive at the simplified version for a single intermediate timestep:

$$\mathcal{L}_{simple} = \mathbb{E}_{t\sim[1,T],x_0\sim q(x_0),\epsilon\sim\mathcal{N}(0,1)}[\|\epsilon - \epsilon_\theta(x_t, t)\|^2]. \tag{8}$$

As stated by Ho et al. (2020) training this setup is beneficial for sampling quality. We refer to the original DDPM paper for more information on the derivations.

## 3. Method

In this section, we demonstrate how our method leverages a single pre-trained DDPM to perform initial anomaly masking (subsection 3.1), stitching (subsection 3.2) and re-sampling (subsection 3.3).

### 3.1. Anomaly Masks

We utilize initial anomaly maps as a guide for our pseudo-healthy (PH) reconstruction. We obtain the anomaly masks from initial reconstructions using high levels of noise.
In contrast to AnoDDPM [1], our method is not as limited by the requirement to preserve intricate healthy details and contextual information, such as specific brain structure features like cortical folding, when generating high-quality pseudo-healthy images.

Instead, we prioritize recall over precision as an optimal setup. Therefore, we favor having more false positives in the anomaly masks to ensure the comprehensive coverage of all anomalies in subsequent steps. Consequently, the limitations and trade-offs associated with AnoDDPM do not apply to our approach. See Section 5.1 and Section 5.2 for more details on the associated trade-offs.

In our study, we empirically set the noise level to $t = 200$, which proved sufficient to remove all anomalies.

We compute the initial mask $\hat{m}$ based on the residual of

the initial reconstruction $\hat{x}_0$ from noise level $t = 200$ and the original image $x$. Since simple residuals tend to be dependent to the underlying pixel intensities, we compute the initial masks based on a combination of absolute and perceptual differences as proposed by Bercea et al. (2023a):

$$\hat{m} = norm_{95}(|\hat{x}_0 - x|) * \mathcal{S}_{lpips}(\hat{x}_0, x), \tag{9}$$

Finally, we dilate the obtained heatmaps $\hat{m}$ with a kernel of 3 to enlarge the margins to the healthy tissues and binarize the masks with a value of 1 marking a possible anomaly.

### 3.2. Stitching

Using the binarized masks $\hat{m}$, we apply two distinct operations. Firstly, utilizing the inverted mask, we generously remove all anomalies from the original anomalous image $(1 - \hat{m}) \odot x$, while preserving its contextual information and underlying brain structure. Secondly, we selectively incorporate portions of the pseudo-healthy reconstruction from the previous part into the image where necessary with $\hat{m} \odot \hat{x}_0^{rec}$. The stitching step noises the cutout original image to step $t$ in every step of the ensuing iteration. The reconstructed counterpart is initialized in the same way using (4) as $x_T^{ph} \sim \mathcal{N}(\sqrt{\bar{\alpha}_t}x, (1 - \bar{\alpha}_t)\mathbf{I})$ with $T = 50$, our starting noise level for the stitching and re-sample process. However, for all subsequent iterations, the reconstructed counterpart is obtained by de-noising the combined result of the stitching from the previous timestep $t$ with (2):

$$x_{t-1}^{context} \sim \mathcal{N}(\sqrt{\bar{\alpha}_t}x, (1 - \bar{\alpha}_t)\mathbf{I}) \tag{10}$$

$$x_{t-1}^{ph} \sim \mathcal{N}(\mu_\theta(x_t, t), \tilde{\beta}_t\mathbf{I}) \tag{11}$$

$$x_{t-1} = (1 - \hat{m}) \odot x_{t-1}^{context} + \hat{m} \odot x_{t-1}^{ph} \tag{12}$$

with the input $x$ conditioning the process with a clearer picture of the context structure in every step.

### 3.3. Re-Sampling

The stitching process of the original image and the initial pseudo-healthy (PH) reconstructed image may result in some regions not aligning perfectly or exhibiting variations in intensity scales, as depicted in the second column of Figure 5. We therefore employ re-sampling for harmonization and in-painting effects. The concept of re-sampling was initially introduced by Lugmayr et al. (2022) for in-painting purposes, which involved pre-defined or manually selected masks. In contrast, our approach leverages automatically generated masks to effectively in-paint anomalous regions and conduct anomaly detection. Additionally, by utilizing a low noise level of $T = 50$ and taking advantage of the valuable information obtained from fitting initial pseudo-healthy reconstructions $\hat{x}_0$ within the cutout anomalous regions of the original image, our method achieves a significant reduc-

---

[1] AnoDDPM encompasses both the specific paper referred to as such (Wyatt et al., 2022) and general methodologies that employ diffusion models in a similar manner

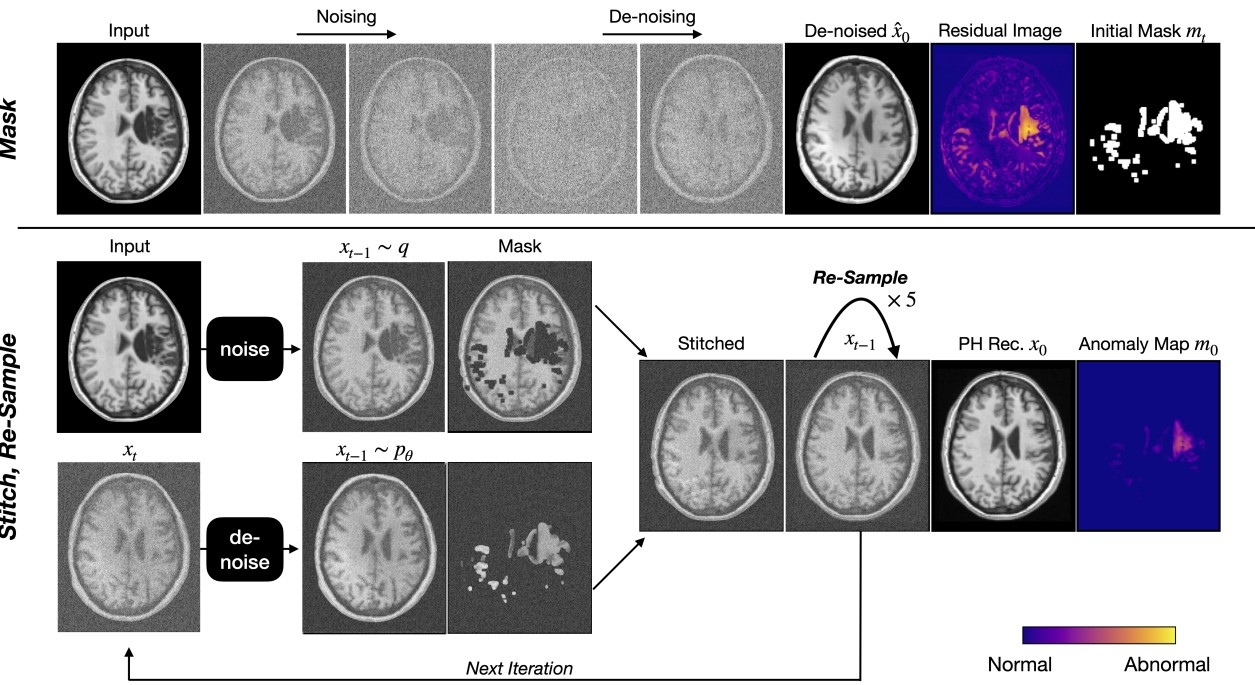

*Figure 2.* Illustration of the proposed *AutoDDPM* method for anomaly detection. Initially, the anomalous input undergoes a diffusion process with a noise level of 200, generating automatic **masks** that represent the initial anomaly maps. The original image and the initial reconstruction are seamlessly **stitched** together. By performing **re-sampling** from their joint distribution, harmonization and in-painting effects are applied, resulting in the final pseudo-healthy (PH) reconstruction and refined anomaly maps. This iterative process effectively replaces anomalous regions while preserving healthy tissues, leading to enhanced robustness and accuracy in anomaly detection.

tion in computational requirements compared to the original approach in (Lugmayr et al., 2022).

This modified strategy describes the process of oscillating between $t$ and $t-1$ for a specified number of re-sample steps. As showed by Lugmayr et al. (2022), separately sampling and putting together the context region and the region to in-paint in every step with a binary cutoff prevents a good harmonization. Thus, to aid the harmonization process, we go back in the reverse process from $t - 1$ to $t$, taking one forward step as described by (1):

$$x_t \sim \mathcal{N}(\sqrt{1 - \beta_t}x_{t-1}, \beta_t\mathbf{I}). \qquad (13)$$

Based on our experimental results, we found that performing 5 re-sampling steps from $t - 1$ to $t$ and vice versa provides satisfactory results. At the end of this process, we obtain the refined pseudo-healthy prediction: $x_0$.

### 3.4. Final Anomaly Maps

The final anomaly maps $m$ can be obtained by directly subtracting the input image from the pseudo-healthy reconstruction. In this work, we aim in utilizing the original uncertainty in the initial estimates $\hat{m}$ to further refine the predictions. We chose this approach based on the motiva-

tion that genuine anomalies would be consistently present in both anomaly maps, while false positives would only appear in one of the predictions. We therefore obtain the final anomaly maps $m$ by multiplying the initial uncertain predictions with the residual anomaly maps: $m = |x_0 - x| * \hat{m}$.

## 4. Experimental Setup

Our experiments are based on training a single AnoDDPM model (Wyatt et al., 2022) and perform smart sampling that includes masking, stitching, and re-sampling to improve it's robustness and accuracy. All proposed techniques do not require further training. To ensure a diverse distribution of the healthy population, we trained the model on two publicly available datasets of healthy brain T1w MRI scans: IXI (ixi), consists of 581 scans, while the second dataset, FastMRI+ (Zhao et al., 2021), comprised 176 scans, with a split of 131 for training, 15 for validation, and 30 for testing.

To assess the performance of the methods, we focused on evaluating the localization of ischemic stroke. For this purpose, we utilized the publicly available ATLAS v2.0 dataset (Liew et al., 2022), which contains 655 images with manually segmented lesion masks created by expert clinicians. To prepare the data for analysis, we normalized the

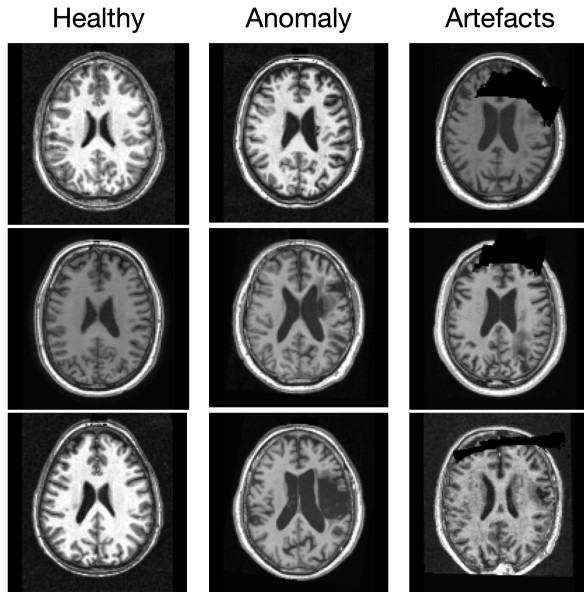

Healthy     Anomaly     Artefacts

*Figure 3.* Samples of healthy, anomaly, and artefacts from the Atlas dataset (Liew et al., 2022).

*Table 1.* **The noise paradox.** As the noise level t increases, the diffusion models effectively remove more anomalies from their reconstructions (AUPRC and $\lceil Dice \rceil$). However, this also leads to an increase in false positives due to inaccuracies in the reconstructions (SSIM). *AutoDDPM* significantly enhances the robustness of diffusion models by achieving reconstructions that are both accurate and free of anomalies. x% shows improvement over best baseline (t=300) and x% shows the decrease in performance compared to *AutoDDPM*.

| Method | PH Rec. SSIM ↑ | Anomaly Segmentation AUPRC ↑ | $\lceil Dice \rceil$ ↑ |
|---|---|---|---|
| AutoDDPM (ours) | **93.41** ▲ 93% | **14.48** ▲ 241% | **22.75** ▲ 200% |
| AnoDDPM (t=50) | 80.10 ▼ 14% | 2.79 ▼ 81% | 5.84 ▼ 74% |
| AnoDDPM (t=100) | 69.14 ▼ 26% | 3.15 ▼ 78% | 6.55 ▼ 71% |
| AnoDDPM (t=150) | 62.10 ▼ 34% | 3.66 ▼ 75% | 7.58 ▼ 67% |
| AnoDDPM (t=200) | 56.26 ▼ 40% | 3.86 ▼ 73% | 7.95 ▼ 65% |
| AnoDDPM (t=250) | 52.38 ▼ 44% | 3.93 ▼ 73% | 7.82 ▼ 65% |
| AnoDDPM (t=300) | 48.39 ▼ 48% | 4.25 ▼ 71% | 8.39 ▼ 63% |

anomaly localization evaluation, we utilized the area under the precision-recall curve (AUPRC) and the maximum Dice score $\lceil Dice \rceil$.

## 5. Results and Analysis

In this section, we conduct a comprehensive evaluation and analysis of our proposed method, shedding light on the limitations of classical diffusion models when applied to anomaly detection. We begin by examining the trade-off between reconstruction quality and anomaly detection in Section 5.1, followed by an in-depth investigation of the performance across different lesion sizes in Section 5.2. Furthermore, we present two ablation studies in Section 5.3 and Section 5.4, which offer valuable insights into the impact of re-sampling and the utilization of inherent uncertainty in the initial mask estimation.

### 5.1. The noise paradox

In this experiment, our focus is to delve into the impact of selecting a noise level during the inference process. This noise paradox revolves around the observation that increasing the noise level would indeed decrease the number of anomalies present in the reconstruction. However, it simultaneously poses a risk of compromising the integrity of the healthy tissues within the image, consequently leading to false positive detections. By thoroughly analyzing this effect, we strive to gain a deeper understanding of the delicate balance between noise levels and the preservation of crucial healthy tissue information. We present the quantitative results in Table 1 and visualize them in Figure 4. AnoDDPM demonstrates a high accuracy of 80.10 SSIM in reconstructing healthy samples from the Atlas dataset at a noise level scale of $t = 50$. However, as the noise

mid-axial slices to the $98th$ percentile, applied padding, and resized them to a resolution of $128 \times 128$.

To ensure the reliability of our analysis, we implemented several pre-processing steps. Firstly, we excluded middle slices that had no visible anomalies (as reported by expert radiologists). We instead, added them as part of the healthy set (N=215). Conversely, slices containing anomalies were labeled as anomalous. By including both healthy and anomalous slices from the same dataset, we aimed to reduce potential confounding effects resulting from domain shifts. Furthermore, during the curation process, we identified certain scans with large hypo-intense imaging artifacts that were not annotated by the radiologists (N=20). Recognizing the potential impact of these artifacts on our performance results, we decided to remove these slices containing such artifacts from the dataset. Figure 3 shows samples from the healthy, anomaly, and artefact distribution.

To evaluate the performance of our method across different lesion sizes, we stratified the test set into three groups based on the size of the lesions. The small group (N=209) comprised the first $25th$ percentile, consisting of lesions smaller than 71 pixels. The large group (N=59) encompassed the top $25th$ percentile, including scans with the largest lesions ($\geq$ 570 pixels). The medium group (N=152) included the remaining scans with lesions of intermediate sizes.

To quantify the accuracy of our reconstructions on healthy scans, we use the structural similarity index (SSIM). For

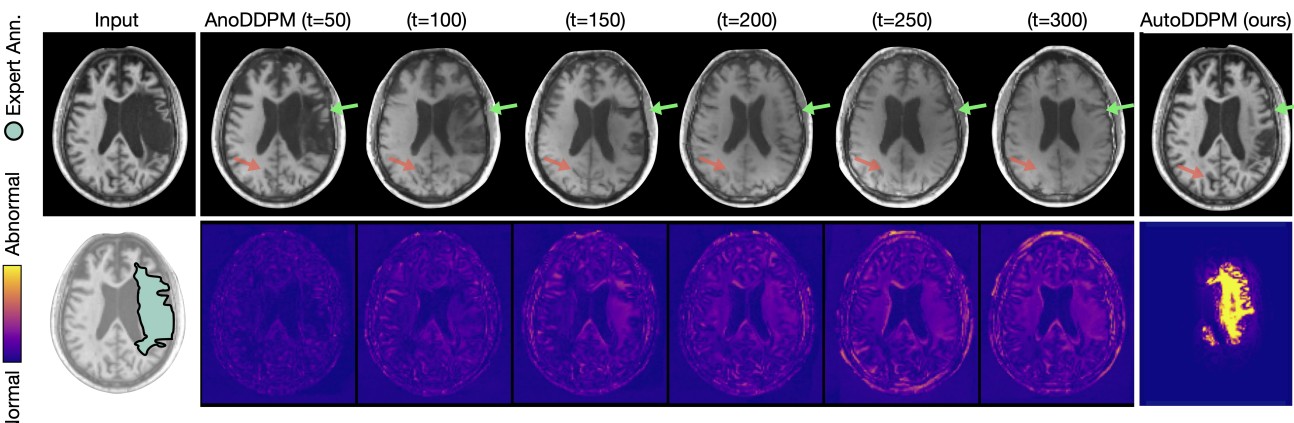

*Figure 4.* **The Noise Paradox.** As the noise level increases, AnoDDPM begins to substitute the lesion with pseudo-healthy tissues (indicated by a green arrow). However, the additional noise disrupts the healthy tissues, resulting in inaccurate reconstructions and false positives (indicated by a salmon arrow). On the other hand, *AutoDDPM* replaces the lesion with pseudo-healthy tissues while successfully preserving the integrity of the healthy tissues. See Table 1 for quantitative evaluations.

level $t$ increases, the reconstruction accuracy of AnoDDPM progressively declines, reaching a significant decrease of 49% and achieving an SSIM of only 48.39 at a noise scale of $t = 300$. In contrast, the anomaly detection accuracy consistently improves with higher noise levels. As depicted in Figure 4, AnoDDPM effectively removes the lesion only at noise levels above $t = 200$. However, the reconstructions already exhibit noticeable deviations in healthy characteristics from the original input, leading to false positive detections. This tradeoff between reconstruction accuracy and detection performance highlights a challenging paradox where no specific operating point can provide optimal solutions. Leveraging these insights, we have developed a refined anomaly detection approach that strikes an optimal equilibrium, ensuring accurate anomaly identification while preserving vital healthy structures. *AutoDDPM* achieves a remarkable reconstruction accuracy of 93.41 SSIM on healthy samples, while also achieving outstanding anomaly detection performance, improving the results of the best performing AnoDDPM by more than 200%.

### 5.2. The unknownness dilemma

In the realm of anomaly detection, researchers have explored various strategies to enhance performance by tailoring the noise distribution to match the anomaly distribution. One notable example is the use of simplex noise, specifically designed to improve the detection of large hypo-intense lesions that mimic tumors in T1w MRI scans. Similarly, varying lesion sizes may necessitate different levels of noise for optimal detection. However, it is important to remember that anomaly algorithms should strive to address the realm of "unknown unknowns" and should not be overly op-

*Table 2.* **The unknownness dilemma.** Optimal noise levels vary for anomalies of different types and sizes. However, since the distribution of anomalies is typically unknown, optimizing the noise level becomes an impractical task. In contrast, AutoDDPM addresses this challenge by enhancing the detection of lesions of various sizes (small, medium, and large) without requiring explicit tuning of the noise levels. x% shows improvement over the best baseline results marked with an underline and x% shows the decrease in performance compared to *AutoDDPM*.

| Method | $\lceil Dice \rceil \uparrow$ | | |
|---|---|---|---|
| | small | medium | large |
| AutoDDPM (ours) | **7.46** ▲ 91% | **23.65** ▲ 145% | **36.77** ▲ 48% |
| AnoDDPM (t=50) | 2.34 ▼ 69% | 6.77 ▼ 71% | 18.06 ▼ 51% |
| AnoDDPM (t=100) | 3.28 ▼ 56% | 8.14 ▼ 66% | 19.78 ▼ 46% |
| AnoDDPM (t=150) | 3.90 ▲ 48% | 9.08 ▼ 62% | 21.32 ▼ 42% |
| AnoDDPM (t=200) | 3.14 ▼ 58% | 9.44 ▼ 60% | 22.35 ▼ 39% |
| AnoDDPM (t=250) | 2.65 ▼ 65% | 9.57 ▼ 60% | 22.38 ▼ 39% |
| AnoDDPM (t=300) | 2.17 ▼ 71% | 9.65 ▼ 59% | 24.83 ▼ 32% |

timized for specific scenarios. In this section, our objective is to thoroughly analyze the performance of AnoDDPM on different lesion sizes and compare it against our proposed method. The numerical evaluation for different lesion sizes is presented in Table 2. It demonstrates that AnoDDPM achieves optimal performance for small lesions at a noise level around $t = 150$, but quickly degrades beyond that point, yielding worse results compared to lower noise levels. Conversely, the detection of medium and large lesions reaches its peak performance at a maximum noise level of $t = 300$. Selecting the ideal operating point becomes a challenge without prior knowledge of the lesion distribution, highlighting the dilemma of unknownness. *AutoDDPM* enhances the detection accuracy for lesions of various sizes,

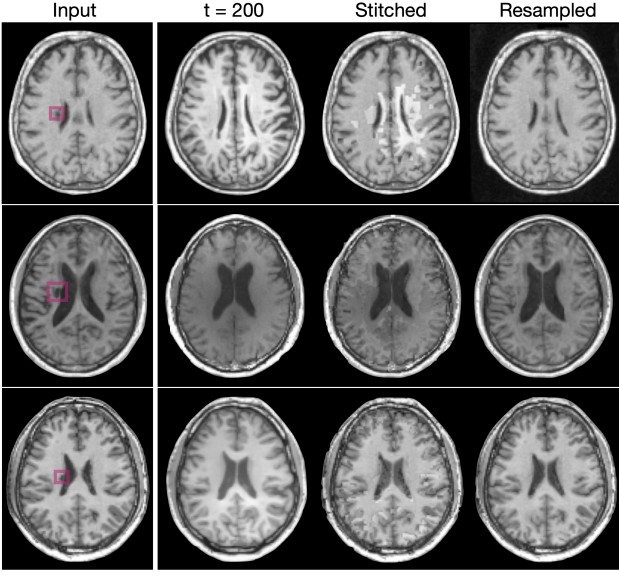

*Figure 5.* **Ablation: Effect of re-sample.** AnoDDPM reconstructions (t=200) are compared to naive stitching of masked inputs and pseudo-healthy reconstructions and to final re-sampled PH reconstructions. The different rows show different samples.

eliminating the need for explicit optimization of the operating point or noise level. Nonetheless, the detection of very small lesions continues to pose significant challenges, with some being nearly imperceptible to the naked eye on low-resolution and 2D slices.

### 5.3. Ablation: Effect of Re-Sampling

The process of re-sampling plays a crucial role in the restoration of anomalous inputs, contributing significantly to refining the pseudo-healthy reconstruction and minimizing the presence of undesired artifacts. Through the application of harmonization and in-painting effects, re-sampling enhances the overall quality and cohesiveness of the stitched images. The absence of this vital step would result in numerous artifacts and inconsistencies when simply stitching the original and pseudo-healthy images together. Qualitative examples, as depicted in Figure 5, highlight the harmonizing effect in the first row, where the initial reconstruction's varying intensities are harmonized to match the target image. In the second row, the re-sampling process not only achieves intensity harmonization but also performs in-painting to introduce new structures into the stitched image, closely resembling the characteristics of the input image.

### 5.4. Ablation: Uncertainty Maps

In our pipeline, the binarization of initial anomaly heatmaps is performed to facilitate the stitching of the original im-

*Table 3.* **Ablation: Effect of uncertainty.**

| Method | $\lceil Dice \rceil$ ↑ | | | |
|---|---|---|---|---|
| | global | small | medium | large |
| AutoDDPM | **22.75** | **7.46** | **23.65** | 36.77 |
| w/o uncertainty | 19.94 | 4.95 | 20.40 | **37.03** |

age and the pseudo-healthy reconstruction. However, this thresholding step results in the loss of the inherent uncertainty contained in the initial estimates. Acknowledging the significance of preserving this uncertainty, we propose to leverage the initial guess and combine it with the final prediction. This proposition is rooted in the observation that the presence of true positives, indicative of actual anomalies, should be evident in both the initial guess and the final prediction. Conversely, false positives, representing false alarms, may be missed in either one. By integrating the initial guess and the final prediction, our objective is to enhance detection performance and capture subtle variations in anomaly localization. The anomaly detection results, as shown in Table 3, demonstrate that incorporating the initial estimates into the final predictions yields an improvement of 14% in detection outcomes. It is especially beneficial for detecting small lesions, where the impact of false positives is much higher, with an increased performance of 51%.

## 6. Discussion

In our study, we shed light on the limitations of diffusion models in anomaly detection and propose a novel method, *AutoDDPM*, to overcome these challenges. The two main limitations we addressed are the noise paradox and the unknownness dilemma.

The noise paradox refers to the trade-off between reconstruction accuracy and anomaly detection performance in diffusion models. As the noise level increases, anomaly detection improves (for medium and large lesions) but at the cost of degraded reconstruction accuracy. Our *AutoDDPM* method strikes a balance between these two objectives, achieving superior performance in both anomaly detection and reconstruction accuracy.

The unknownness dilemma arises from the difficulty of choosing the optimal operating point or noise level for anomaly detection, especially when dealing with lesions of different types or sizes. Our proposed method improves the detection of lesions of various sizes by a large margin without the need to optimize for the noise level.

While our method demonstrates promising results, it is important to acknowledge its limitations. The detection of small lesions remains challenging, especially in low-resolution and 2D slices. To address this, we aim to explore and develop advanced techniques in future research, poten-

tially incorporating additional modalities and extending our networks to process 3D volumes.

In our early attempts, we have recognized the challenge of preserving and utilizing the inherent uncertainty during the binarization process in our approach. By just multiplying the initial uncertain predictions with the final anomaly maps, we were able to improve the anomaly detection results, especially for small lesions. We believe that capturing and utilizing uncertainty can significantly enhance the accuracy and robustness of anomaly detection. As part of our future work, we plan to further investigate and develop techniques that effectively preserve and leverage the inherent uncertainty information.

The smart sampling method presented in this paper significantly enhances performance without requiring re-training or introducing additional complexity. However, it is important to note that the additional diffusion steps for resampling may slightly increase inference times. Nonetheless, since this step involves de-noising from very low noise levels, such as $T = 50$, the impact on inference time is minimal. Additionally, our proposed method is independent of general advancements in sampling techniques for diffusion models and can benefit from different sampling schemes that further decrease inference times.

AnoDDPM suffers from the noise paradox, where no operating point (noise level) can optimize both the removal of anomalies and preservation of healthy tissues. While we address the need for precise noise level selection and offer a solution to the noise paradox, we are still required to determine an initial and refinement noise level. Although the general idea is to select high levels for the initial step and low levels for the refinement step, we plan to investigate in our future research how these parameters can be automatically derived for each individual input scan.

In conclusion, our study contributes to the field of anomaly detection by highlighting the limitations of diffusion models and proposing a novel approach, *AutoDDPM*, to overcome these challenges. We believe that our findings improve the robustness and interpretability of diffuson models and open up avenues for further research and advancements in anomaly detection in medical imaging.

## 7. Acknowledgements

C.I.B. is in part supported by the Helmholtz Association under the joint research school "Munich School for Data Science - MUDS".

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
