# OpenReview forum: "Mask, Stitch, and Re-Sample: Enhancing Robustness and Generalizability in Anomaly Detection through Automatic Diffusion Models"
_ICML.cc/2023/Workshop/IMLH — IMLH 2023 Poster_

### Official Review · Reviewer_qqqU · 2023-06-11
**Comments on IMLH 2023, Submission 65**

**Rating:** 7
**Confidence:** 4

**Review:**

In this paper, the author introduces AutoDDPM, a novel approach designed to enhance the robustness of diffusion models. The proposed method effectively addresses two main limitations: the noise paradox and the unknownness dilemma. The paper provides a detailed explanation of the method and presents experimental results to demonstrate its effectiveness. The results clearly indicate that the proposed method outperforms previous models. However, there is room for improvement in the paper's expression, as some statements can be confusing. For instance, the paper mentions the utilization of the AnoDDPM method (Wyatt et al., 2022) in the experiments, which may give the impression that the author simply applied AnoDDPM to this research.

---

### Official Review · Reviewer_5y8V · 2023-06-18
**Effective pipeline with minor issues.**

**Rating:** 9
**Confidence:** 5

**Review:**

The authors have introduced an algorithmic framework, AutoDDPM, which aims to identify and reconstruct pathological anomalies. This pipeline employs pre-trained diffusion models and encompasses three essential stages: masking, stitching, and resampling. The masking stage involves a comparison between the original input and the reconstructed image, while the stitching stage aims to integrate additional original context. The final step, resampling, seeks to mitigate inconsistencies and artifacts arising from the stitching process. This methodology exhibits a marginal improvement in reconstruction score and segmentation accuracy compared to the preceding AnoDDPM baseline, and offers the potential advantage of obviating the need for meticulous selection of an appropriate noise level.

Strengths:

This paper is overall well written, with clear explanations provided for the ideas, methods, and experiments. The significance of this work lies in the amalgamation of masking, stitching, and resampling techniques to establish an effective pipeline. Notably, the simplicity of the proposed approach is evident in its reliance on a single pretrained diffusion model, while other operations are performed during the inference stage. The observed improvement achieved by the proposed pipeline, AutoDDPM, when compared to the baseline model AnoDDPM, is indeed noteworthy. This improvement underscores the effectiveness of the pipeline in addressing the limitations of diffusion models. Moreover, the practical applicability of the AutoDDPM pipeline appears promising, suggesting its potential suitability for real-world deployment.



Weaknesses:

1. The authors extensively discussed the noise paradox and provided a comprehensive analysis of the previous model, AnoDDPM, highlighting the challenges associated with finding an optimal noise level for all anomalies. However, it is noteworthy that a similar analysis was not conducted on their proposed model, AutoDDPM. The absence of comparable experiments in the evaluation of AutoDDPM limits the ability to definitively conclude that this model has effectively addressed the noise paradox. Conducting a similar set of experiments and analyses on AutoDDPM would provide a more robust assessment of its ability to handle the noise paradox and enhance the credibility of the proposed solution.

2. Although the authors put forward a direct pipeline consisting of several essential components for optimal performance, the inference procedure appears to be relatively lengthy. While the necessity of each component is acknowledged, there is a concern regarding the running time during inference. It remains unclear whether the five resampling steps are performed at each iteration for T=50 times or solely once initially. If it is the former case, it raises questions regarding its potential impact on the overall inference time.

Conclusion:

In conclusion, the paper presents a valuable contribution to the field by proposing an efficient and promising pipeline. However, more analysis should be done in order to conclude that the method has truly addressed the noise paradox. Furthermore, reporting the inference time would provide valuable insights into the practical applicability of the pipeline, demonstrating its feasibility in real-world scenarios. Addressing these aspects would strengthen the overall impact and credibility of the proposed approach.

---

### Meta-Review · Area_Chair_Rjdz · 2023-06-20

**Recommendation:** Accept (Poster)
**Confidence:** 5

**Metareview:**

Both reviewers expressed positive opinions about this paper, appreciating its clarity and extensive experimentation. I kindly ask the authors to carefully consider the identified shortcomings and ensure that these issues are addressed in the final version.

---

### Decision · Program_Chairs · 2023-06-20

Accept (Poster)